# Impact of a collaborative model on community clinician confidence in child and adolescent mental health care, wellbeing, and access to child psychiatry expertise

Elise D'Abaco[1]©, Sonia Khano[2,3]©, Al Giles-Kaye[2,4], Jag Dhaliwal[5‡], Ric Haslam[6‡], Chidambaram Prakash[6‡], Harriet Hiscock[1,2,3,7]©*

1 Centre for Community Child Health, Royal Children's Hospital Parkville, Melbourne, Victoria, Australia, 2 Murdoch Children's Research Institute, Health Services, Parkville, Melbourne, Victoria, Australia, 3 Health Services Research Unit, Royal Children's Hospital Parkville, Melbourne, Victoria, Australia, 4 Melbourne Graduate School of Education, University of Melbourne Parkville, Melbourne, Victoria, Australia, 5 North Western Melbourne Primary Health Network, Melbourne, Victoria, Australia, 6 Mental Health, Royal Children's Hospital Parkville, Melbourne, Victoria, Australia, 7 Department of Paediatrics, The University of Melbourne, Parkville, Melbourne, Victoria, Australia

© These authors contributed equally to this work.
‡ These authors also contributed equally to this work.
* harriet.hiscock@rch.org.au

**Data Availability Statement:** All relevant data are now available in a public repository with the following name and link: File name: COMPASS

## Abstract

### Background

The COVID-19 pandemic was associated with an increase in child and adolescent mental health disorders, with subsequent worsening of patient access to specialist mental health care. Clinicians working in the community were faced with increased demands to diagnose and manage pediatric mental health disorders, without always having the confidence and knowledge to do so. We therefore developed COnnecting Mental-health PAediatric Specialists and community Services (COMPASS)—a collaborative model designed to upskill community clinicians in child and adolescent mental health care and provide them with better access to child and adolescent psychiatry expertise. COMPASS comprises (1) an online Community of Practice (CoP) with fortnightly one-hour sessions covering: anxiety; aggression and challenging behaviours; depression; self-harm and suicidality; eating disorders; and autism spectrum disorder/complex cases and (2) primary and secondary consultations for general practitioners and paediatricians with an experienced child psychiatrist. We aimed to assess the impact of COMPASS on community clinician self-reported confidence in: managing common child and adolescent mental health disorders (Objective 1, primary outcome); navigating the mental health care system (i.e. knowing how services are organised, accessed, and how to refer patients, Objective 2); diagnosing conditions (Objective 3); prescribing psychotropic medications (Objective 4) as well as the impact on, clinician practice and wellbeing (Objective 5) and outcomes of patients referred by COMPASS clinicians to the child psychiatrist (Objective 6).

survey data- 22-08-24 https://figshare.com/s/a0ecdf991e56bbffa8a4.

**Funding:** The author(s) received no specific funding for this work.

**Competing interests:** The authors have declared that no competing interests exist.

## Methods

We evaluated COMPASS in its first year, with COMPASS running from March to July 2021. Participating clinicians completed pre-post surveys evaluating change in Objectives 1 to 4 above, using study-designed measures. A purposive sample of clinicians was then invited to a semi-structured interview to understand their experience of COMPASS and its impacts on practice and wellbeing (Objective 5). We adopted an inductive approach to the qualitative analysis using the Framework Method. This involved selecting five random transcripts which were double coded and categorized, to generate an initial framework against which all subsequent transcripts were analysed. Themes and subthemes were generated from the data set, by reviewing the matrix and making connections within and between clinicians, codes and categories One child psychiatrist completed a 2-week logbook of the nature and outcomes of primary and secondary consultations (Objective 6).

## Findings

51 (86%) clinicians attended CoP sessions and completed pre-post surveys, with 92% recommending COMPASS to peers. Clinicians reported increased confidence in the pharmacological and non-pharmacological management of all conditions, most notably for management of self-harm. They also reported increased knowledge of how to navigate the mental health system and prescribe medications. Qualitative analysis (n = 27 interviews) found that COMPASS increased clinician wellbeing and reduced feelings of professional isolation and burnout. Over the 2-week snapshot, the child psychiatrist consulted on 22 patients and referred all back to the community clinician.

## Conclusions

COMPASS is associated with improved clinician confidence to manage child and adolescent mental health concerns, navigate the mental health system, improved clinician wellbeing, and reduced need for ongoing mental health care by specialists.

## Introduction

Mental disorders are a leading cause of morbidity in children and young people, worldwide [1]. One in seven 10-19-year-olds experience a mental health disorder, accounting for 13% of the global burden of disease in this age group [1]. Similarly, in Australia, 14% of children aged 4–17 years of age meet the criteria for a mental health disorder over a 12-month period [2,3]. Mental health disorders are associated with adverse impacts on child function and wellbeing, including reduced social engagement [4], school refusal [5] and educational under-achievement [6]. Left untreated, disorders can persist into adulthood [6,7]. As such, childhood provides an important opportunity for early intervention.

To provide early intervention, a healthcare system needs to have services that are accessible to children and a workforce that is suitably trained to provide pediatric mental health care and navigate access to specialist care when required. This is not the status in Australia. Two recent inquiries—the Royal Commission into Victoria's Mental Health System and the National Mental Health Commission—identified systemic issues impacting access to mental health care for children and adolescents [8,9]. These include a lack of service capacity, inequitable access

to care (including reduced access in rural areas) and limited focus on the early years, meaning that young children often miss out on the care they need [10]. Further, system access is increasingly driven by crisis, with emergency departments being used as entry points [11]. The COVID-19 pandemic has exacerbated this issue [12].

The provision of clinical care for mild to moderate child and adolescent mental health conditions in Australia sits primarily with community-based clinicians [2]. General practitioners (GPs) are the most accessed clinicians for mental health presentations (35%), followed by psychologists (23.9%), paediatricians (21%) and counsellors (20.7%) [2]. However, many community clinicians report they have inadequate training in pediatric mental health and require more professional development opportunities to improve their confidence in diagnosing and managing child and adolescent mental health conditions and to navigate the complex mental health system [13]. We previously interviewed 143 child and adolescent psychiatrists, pediatricians, child psychologists and general practitioners to understand their perspectives on problems and solutions for the Australian pediatric mental health system. Clinicians reported problems with service fragmentation, long wait times, and for GPs and pediatricians—inadequate training in child and adolescent mental health [13]. Solutions included increased access to child psychiatry expertise, mental health training, and co-located multidisciplinary services. A Community of Practice (CoP)—whereby community-based clinicians learn from an expert–could operationalise some of these proposed solutions by increasing access to child psychiatry (the expert) and providing mental health training and knowledge about how to navigate the system. CoPs are "groups of people who share a concern or a passion for something they do and learn how to do it better as they interact regularly" [14]. CoPs can increase productivity, clinician connectedness and provide evidence-based practice [15–18]. Other models to improve access to child psychiatry expertise and provide mental health training include primary and secondary specialist mental health consultations to referring clinicians with the aim of providing timely assessments and advice and return of the client to the referring clinician where possible. The Massachusetts Child Psychiatry Access Program (MCPAP) is an example of such a model [19–22]. It operates at a regional level, providing access (telephone, face-to-face) to child psychiatrists for primary care pediatricians as well as education modules for common mental health conditions. It has been shown to increase primary care providers self-reported ability to meet the needs of psychiatric patients from 8% to 63% [22].

However, few such models exist in Australia and none, to our knowledge, have been evaluated. Further, it is unknown whether such models impact clinician wellbeing or if they are associated with a reduction in referrals to specialist services. Co-designed with community clinicians in January 2021, this study aimed to expand on previous research by implementing and evaluating a model to improve community clinicians' confidence in the diagnosis and management of common child and adolescent mental health conditions, how to navigate the system, and increase their access child psychiatry expertise, on a background of increasing demand during the COVID-19 pandemic. The *COnnecting Mental-health PAediatric Specialists and community Services (COMPASS)* collaborative model consists of a multi-disciplinary online Community of Practice (CoP) and primary and secondary consultations for referring GPs and paediatricians with experienced child psychiatrists. COMPASS was developed in partnership with the regional primary care network, local clinicians, and the local child and adolescent mental health service. Now in its third year, the primary care network provides funding for the CoP administration (participant registration, conduct of the CoP sessions, emailing of resources to participants after each session), with the mental health service providing funding for the child psychiatrist.

Using clinician pre-post surveys, in-depth interviews and child psychiatrist consultation data, we aimed to determine whether COMPASS could increase clinician self-reported confidence in: managing common child and adolescent mental health disorders (Objective 1, primary outcome); navigating the mental health care system (i.e. knowing how services are organised, accessed, and how to refer patients, Objective 2); diagnosing conditions (Objective 3), prescribing psychotropic medications (Objective 4) as well as the impact on clinician practice and wellbeing (Objective 5) and outcomes of patients referred by COMPASS clinicians to the child psychiatrist (Objective 6). If effective, COMPASS could offer an acceptable way to improve capacity of community clinicians to provide pediatric mental health care, navigate the mental health system and improve their access to child psychiatry expertise. This in turn could reduce the burden on overloaded specialist mental health services and provide policy makers with scalable solutions to improve workforce capability.

## Materials and methods

### Recruitment of participants

Recruitment occurred through a partnership with the North Western Melbourne Primary Health Network (NWMPHN), in the state of Victoria, Australia. Local GPs, paediatricians, and Mental Health (MH) clinicians from the metropolitan area of the NWMPHN were recruited through a NWMPHN broadcast email and personal emails from the senior author (a paediatrician). Participants were eligible to take part in the COMPASS model if they were working in the NWMPHN region and providing care to paediatric patients (0–18 years) with mental health concerns. We received interest in participating from 80 clinicians, of whom 75 were eligible and formally invited to take part in COMPASS.

### Co-development of the CoP

Clinicians attended up to two, one-hour co-development sessions in February 2021 to develop the content and structure of the CoP sessions. The first session focused on the key conditions clinicians wanted training in and how, with clinicians selecting common conditions that they saw in their clinical practice and opting for expert summaries on diagnosis and management followed by a case discussion. The second session focussed on how the CoP would run e.g. duration of the CoP sessions, when to conduct them (i.e. choice of during and after hours), and preferred virtual conferencing platform.

### Delivery of the CoP sessions

Over a five-month period (March to July 2021), 10 fortnightly CoP sessions were delivered to participating community clinicians via an online videoconferencing platform (Zoom©). CoP sessions focused on five key areas: 1) anxiety; 2) aggressive and challenging behaviours; 3) depression, suicidal ideation and self-harm; 4) eating disorders; and 5) complex MH disorders (e.g. autism spectrum disorder). Each session included a didactic teaching lecture on the assessment and then the management of the condition(s) led by the senior child psychiatrist, followed by case-based discussions led by a general practitioner facilitator. After each session, the NWMPHN collated and shared resources with clinicians including screening and assessment tools and evidence-based treatment guidelines. Participating clinicians were encouraged to submit a case study for each session, to discuss aspects of assessment, referral, and management with the multidisciplinary group. Fifty-nine of the 75 eligible clinicians who registered to take part in the CoP attended at least one session, with 60% of clinicians attending more than 5 of 10 sessions (see S1 Appendix).

## Child psychiatry consultation service

In addition to the CoP sessions, the child psychiatrist provided further support to participating clinicians through a consultation service, funded by the Royal Children's Hospital, Melbourne. The child psychiatrist was available by phone, email, or clinic appointments to provide GPs and paediatricians with medication advice, diagnostic, assessment, management, or referral options for their patients. The service was promoted via the NWMPHN network communications including their website and newsletters.

## Data collection

To assess Objectives 1 to 4, we conducted pre- (see S2 Appendix) and post-CoP (see S3 Appendix) online surveys. Clinicians who expressed interest in completing an interview in the pre-survey and attended two or more CoP sessions, were eligible for interview (see below). Interviews aimed to understand clinicians' perceptions of COMPASS and its impact on their well-being (Objective 5). We chose qualitative interviews as a methodology to gain a richer, more in-depth understanding of clinician perspectives than that provided by survey data.

The child psychiatrist completed a study-designed logbook to record the nature and outcomes of their consultations over a two-week period (Objective 6).

## Pre-post surveys

Clinicians were sent pre and post online surveys with two reminders through REDCap, a secure, web-based application for building and managing online surveys and databases, developed by Vanderbilt University [23,24]. All clinicians were sent the pre survey one month prior to the commencement of the CoP and the post survey one week following the last CoP session. Pre- and post-surveys included demographic items (pre-survey only) including clinician age, gender, years of practice etc (see Table 1) and confidence in pharmacological and non-pharmacological management of the specific conditions covered in the CoP (Objective 1). Surveys also asked about knowledge of how mental health services are organised and how to access and refer to them (Objective 2), overall confidence in diagnosing mental health conditions (Objective 3), and confidence (where relevant) in prescribing first line and second/third line psychotropic medications (Objective 4). For all these variables, clinicians were asked to respond separately for child versus adolescent mental health as we hypothesised that clinicians would be less familiar and confident with child versus adolescent mental health care.

Clinicians were asked to rate their responses on a study-designed 4-point Likert scale we have previously used with GPs [23], ranging from *"not at all confident"*, *"not very confident"*, *"fairly confident"*, to *"completely confident"*. Clinicians could also nominate *"not part of my role"* (e.g. for psychologists who do not prescribe medications). *Confidence options* were then grouped into two categories: *"not confident"* (comprising *"not at all confident"* and *"not very confident"* responses) versus *"confident"* (comprising *"fairly confident"* and *"completely confident"* responses), as per our previous research [25]. The post survey also asked clinicians to rate their feedback of the model and whether they would recommend it to peers.

We used simple statistics (e.g. proportions, means, standard deviations) to describe clinician sample demographics. A McNemar test, with alpha set at 0.01 to allow for multi comparisons, was performed to calculate the change in dichotomised confidence scores in the relevant variables for children and adolescents separately, across pre and post time points. We considered changes with a p value of 0.05 or less to be statistically significant. Where applicable, responses to "not part of my role" were excluded from analysis. All quantitative analyses were conducted in Stata 16 (StataCorp LLC, College Station, TX, USA).

**Table 1. Demographic characteristics of clinician participants (n = 59).**

| Characteristic | Number, % |
|---|---|
| Clinician gender | |
| Female | 50 (84.7) |
| Male | 9 (15.3) |
| Clinician role | |
| General Practitioner | 19 (32.2) |
| Psychologist | 19 (32.2) |
| Paediatrician | 11 (18.6) |
| Social Worker | 4 (6.8) |
| Mental Health Nurse/Clinician | 4 (6.8) |
| Occupational Therapist | 2 (3.4) |
| Duration of practice | |
| Less than 6 years | 18 (30.5) |
| 6 to 15 years | 21 (35.6) |
| More than 15 years | 20 (33.9) |
| Number of clinical sessions per week | |
| Less than 6 | 16 (27.1) |
| 6 to 10 | 32 (54.2) |
| More than 10 | 11 (18.7) |
| Paediatric patients seen per week | |
| Less than 11 | 28 (47.5) |
| 11 to 20 | 18 (30.5) |
| More than 20 | 13 (22) |
| Formal training in Paediatric Mental Health | |
| Yes | 18 (30.5) |
| No | 41 (69.5) |

## Clinician interviews

To address Objective 5, eligible clinicians were invited to take part in a qualitative interview with an independent researcher (ED). Interested clinicians were sent an email containing the participant information form (S4 Appendix) and an online link to book a telephone interview time. The researcher telephoned participants to explain the study, confirm eligibility, obtain verbal consent, and conduct the interview. Qualitative interviews were undertaken with the support of an interview guide (S5 Appendix) that allowed flexibility to explore participants experiences and perspectives. Interviews were audio recorded and transcribed verbatim by a secure online transcription service (Rev©) and cleaned by researchers ED and AG-K. Transcripts were then coded for analysis within NVivo version 15 Analysis Software (QSR International Pty Ltd., Cardigan, UK). Clinicians were offered the opportunity to review their interview transcript, however none did so.

We adopted an inductive approach to the qualitative data analysis, using the Framework Method by Gale et al. [26] which sits within the broader methodological framework of content analysis [27]. Content analysis was chosen to systematically organise data into a structured matrix, which addressed the key aims of the study. Themes and subthemes were generated from the data set by reviewing the matrix and making connections within and between participants, categories, and codes. All transcripts were coded to the framework matrix, although data saturation was reached after coding approximately half of the transcripts. E.D and A. G-K. undertook fortnightly meetings to resolve coding discrepancies and track coding

decisions. They also completed journaling activities during the data collection and analysis phase to reflect on personal biases and preconceptions, as well as their relationship to participants, and how this may impact the research outcomes. Additional strategies to ensure rigor of the qualitative analysis, such as member checking and triangulation, were not able to be completed due to time constraints of the study.

### Child psychiatrist logbook

To address Objective 6, the child psychiatrist maintained a logbook over a two-week period, recording the nature of the primary and secondary consultations delivered and resulting outcomes (S6 Appendix). We summarise these findings using descriptive statistics.

### Ethical aspects

Ethics approval was granted by the Royal Children's Hospital Human Research Ethics Committee (HREC 73998/ QA 73663). Due to the low-risk nature of the surveys, participants' completion of the surveys was approved as implied consent. To ensure confidentiality, any information that connected participant contact details to their online survey was stored securely and separately on a password protected computer and only accessed by SK.

During qualitative interviews, informed consent was sought whereby participants were informed that their participation was voluntary and unpaid and verbal consent was sought over the phone by ED. Several steps were taken to ensure confidentiality of participants. Identifying details were removed from the recordings and transcripts, and participants were given an ID number. Electronic data, including interview recordings and transcripts, were stored on the network drive on a password protected computer. At the conclusion of the study, all identifying participant data was destroyed and final study data was stored in a non-identifiable format.

## Results

### Online community of practice (CoP)

**Sample characteristics.**   Fifty-nine participants (19 GPs; 19 psychologists 11 paediatricians; 4 MH nurses; 4 social workers; and 2 MH occupational therapists) registered their interest to take part and attended at least one CoP session and 51 (86%) participants completed both pre and post surveys. Table 1 provides a summary of baseline sample characteristics.

**Objective 1: Managing mental health conditions.**   Overall, clinicians reported increased confidence in non-pharmacological (Fig 1) and pharmacological (Fig 2) management of MH conditions in both children and adolescents. Increases were greatest for non-pharmacological management of self-harm (children: 35.6% to 74.5%; adolescents: 52.5% to 76.5%) and suicidal ideation (children: 32.2% to 58.8%; adolescents: 45.8 to 68.6%), respectively.

### Objectives 2, 3 and 4: Confidence in navigating the mental health system, diagnosing conditions, and prescribing psychotropic medication

For children more so than adolescents, clinicians reported improved confidence in knowing how MH services are organised, how to access services, how to diagnose problems, refer to services, and where relevant, prescribe psychotropic medication (see Table 2). However, post COMPASS, only one half to a third of clinicians felt confident in prescribing first-line (increase from 32.2% to 53%) and second/third line psychotropic medications (13.6% to 37.2%), respectively.

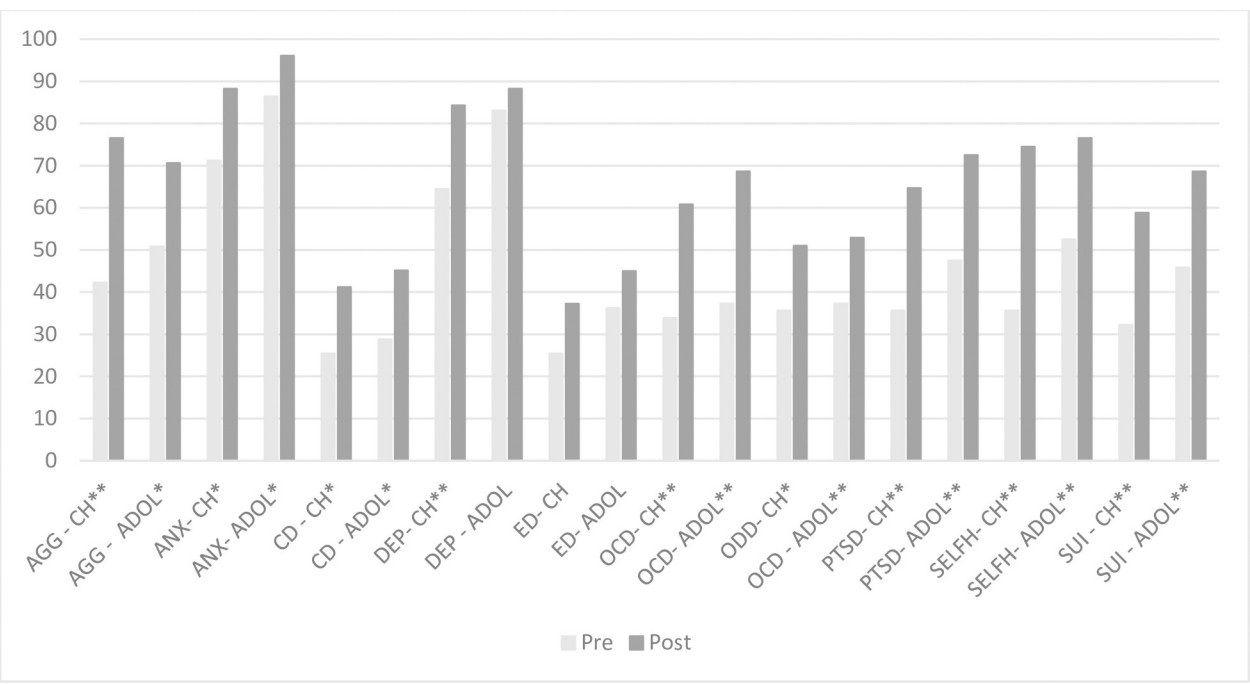

**Fig 1. Change in clinician reported confidence in non-pharmacological management of MH conditions for children and adolescents (n = 51).**
Note, * = <0.05, ** = <0.001. CH = Children; ADOL = Adolescents; AGG = Aggression/challenging behaviours; ANX = Anxiety; CD = Conduct disorder; DEP = Depression; ED = Eating disorder; OCD = Obsessive compulsive disorder; ODD = Oppositional defiant disorder; PTSD = Post-traumatic stress disorder; SELFH = Self-harm; SUI = Suicidality.

## Objective 5: Understanding impact of COMPASS on clinician practice and wellbeing

**Sample characteristics for qualitative interviews.**   Overall, 27 eligible clinicians from the 59 clinicians who attended the CoP sessions expressed interest in completing an interview of whom one declined to be involved and five failed to book an interview time despite three reminder notifications. Reasons for non-participation were not ascertained. Twenty-one interviews were completed. Interviews lasted between 12 to 48 minutes. The sample included 5 GPs, 4 paediatricians, 12 psychologists and mental health workers; most were female (n = 18). Clinicians had high rates of attendance at the CoP, (40% attending 8 or more sessions) and were clinically experienced with 42% having worked for more than 15 years.

**Summary of themes and subthemes.**   Analysis of the interview transcripts generated important themes presented below with representative quotes (Table 3). Quotes have been truncated where necessary without changing the meaning. This is represented by an ellipsis. Words which have been added to quotes by the researchers, to clarify meaning, are contained in square brackets.

## Theme 1: Experience of the CoP and areas for improvement

**Subtheme: Program structure, content and delivery.**   The structure of sessions, including both didactic teaching and case-based discussion, was generally well received by clinicians from all professional backgrounds. Clinicians also valued access to tertiary level expertise through secondary consultation. Case presentations were found to be intellectually stimulating and generated collaborative discussion. However, some clinicians felt case presentations were under-utilized or were overly complex, which narrowed the discussion. Topics covered in the

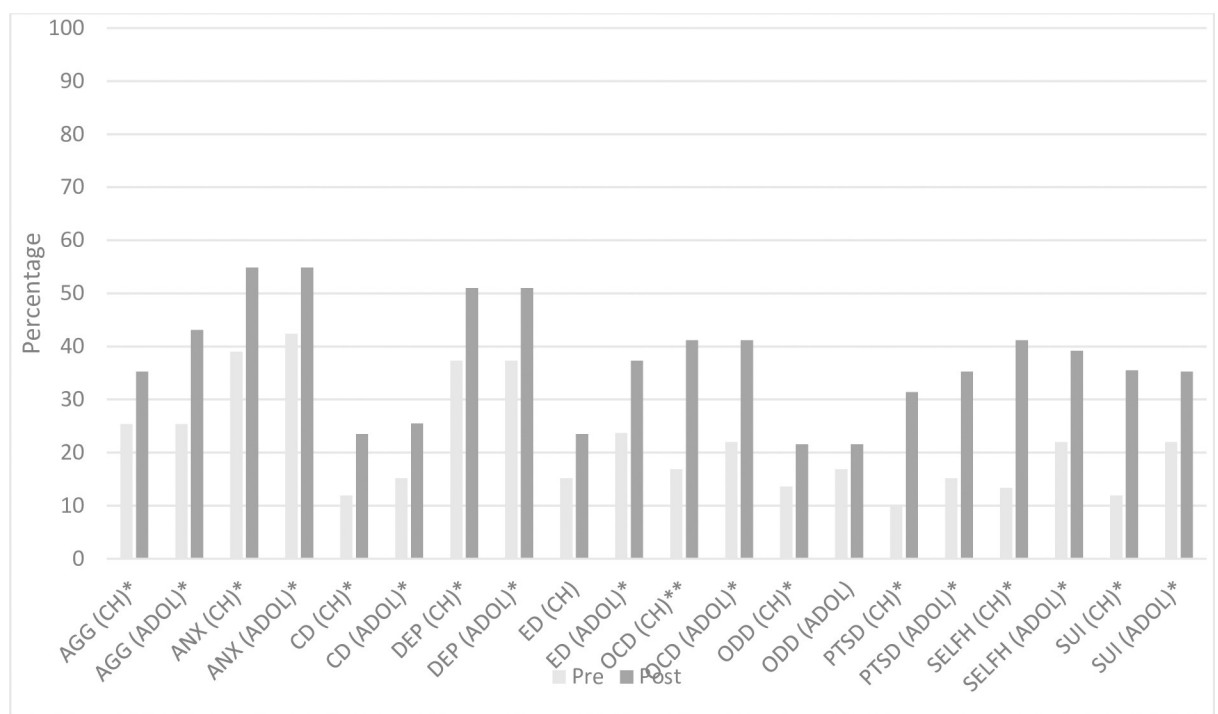

**Fig 2. Change in clinician reported confidence in pharmacological management of MH conditions for children and adolescents (n = 31).** Note, analyses only include clinicians who reported prescribing as part of their professional role * = <0.05, ** = <0.001. CH = Children; ADOL = Adolescents; AGG = Aggression/challenging behaviours; ANX = Anxiety; CD = Conduct disorder; DEP = Depression; ED = Eating disorder; OCD = Obsessive compulsive disorder; ODD = Oppositional defiant disorder; PTSD = Post-traumatic stress disorder; SELFH = Self-harm; SUI = Suicidality.

sessions were described as clinically relevant, particularly the presentation on eating disorders, which was felt to achieve a good balance between medical and psychological models of care. Teaching on psychopharmacology was interesting for those with both medical and non-medical backgrounds. However, a recurrent theme in the analysis was that the content had an over-emphasis on a medical approach to assessment and management of mental health conditions. Clinicians also requested more content on psychological therapeutic models of care, and avenues to support families in crisis.

**Table 2. Change in clinician confidence in knowing how to navigate mental health services, diagnose, and prescribe psychotropic medications.** Note, where applicable, responses to "*not part of my role*" were excluded from analysis.

|  | Children | | | Adolescents | | |
|---|---|---|---|---|---|---|
|  | **Pre**<br>**n = 59 (%)** | **Post**<br>**n = 51 (%)** | **P value** | **Pre**<br>**n = 59 (%)** | **Post**<br>**n = 51 (%)** | **P value** |
| How mental health services are **organised** | 40 (67.8) | 41 (83.3) | 0.07 | 50 (84.7) | 46 (90.2) | 0.36 |
| How to **access** mental health services | 44 (74.6) | 46 (90.2) | 0.02 | 50 (84.7) | 47 (92.2) | 0.20 |
| How to **diagnose** a mental health condition | 40 (67.8) | 40 (78.4) | 0.31 | 42 (71.2) | 43 (84.3) | 0.03 |
| How to **refer** for mental health support | 41 (69.5) | 46 (90.2) | 0.002 | 50 (84.7) | 46 (90.2) | 0.20 |
| **Prescribe first-line** psychotropic medications | 19 (32.2) | 27 (45.7) | 0.001 | 26 (44.1) | 29 (49.1) | 0.04 |
| **Prescribe second and third-line** psychotropic medications | 8 (13.6) | 19 (32.2) | 0.001 | 11 (18.6) | 20 (33.9) | 0.006 |

**Table 3. Quotes representing theme 1: Experience of the CoP and areas for improvement.**

| Sub-Theme | Experience of the CoP | Areas for Improvement | Participant Derived Solutions | Illustrative Quotes |
|---|---|---|---|---|
| Program Structure, Content and Delivery | *Structure* Access to psychiatry expertise and consultation liaison service | More opportunities for reflective practice | Use of structured teaching cases | "... it's good to have a child psychiatrist there because you know, often it's really difficult to access specialists... that's been really helpful" (Mental Health Social Worker 2) "I think that the teaching cases would have facilitated a lot more discussion... [rather than] de novo cases... [using] two or three questions that were common across every session, that draw out the kind of critical pathways in care and management of a child and their family... using pre-prepared case studies to sort of illustrate that" (Psychologist 18) |
|  | *Content* Lectures on eating disorders interesting and clinically relevant | Over-emphasis on pharmacological treatment | More content on psychological therapeutic models, referral avenues for families in crisis | "I'm a psychologist... not a medic...a lot of the educational content was very geared towards medical colleagues who, you know, are in the business of deciding what and when to prescribe" (Psychologist 18) "... it would have been nice had it been more around the combination of therapy and medication... there weren't any new therapeutic models discussed" (Mental Health Social Worker 3) |
|  | *Delivery* Online format convenient and enabled regular attendance Resources provided were extremely useful | Online format a barrier to group discussion and cohesion | Offer virtual and in person options Clear rules for online engagement | "I know we are supposed to be there with our big ears on and (laughter) watching intently, but you know, at 6.30pm with three kids, it's a bit tricky..." (General Practitioner 11) "I've been doing Telehealth during lockdown, so you get a little bit desensitized and check out a little bit" (Psychologist 14) "... the content is constantly evolving... so to get that kind of information all put together at the end of the webinars was really good... because there is some stuff that I probably wouldn't have come across in my day-to-day work" (Paediatrician 17) |
| Multi-Disciplinary Model | Enabled professional connections Insight into skill set and knowledge base of other clinicians | Address factors impacting group dynamics (i.e., group size, professional mix, clear rules for online engagement) | Facilitation to actively manage group dynamics Smaller online groups Clinician specific breakout groups | "... I'm a psychologist... so being able to hear from say paediatricians and GPs and other people who work with the same client group... but in different ways... I found that really interesting" (Psychologist 12) "... people who already work with a lot of mental health issues, have a fairly good understanding around DSM criteria... the tricky thing was to understand where those gaps in knowledge were, especially for GP... being a mixed group...' (Paediatrician 17) "... so, in our session, there were a couple of very dominant voices. Um, and I just got the sense sometimes that there are people with some such expertise in that room that we weren't hearing from..." (Psychologist 18) "[It was] unclear whether to... put a comment in the chat room... whether it'll be picked up there or wave your hand or, um, or just jump in... probably those housekeeping rules need to be mentioned each time? Like, you know... please put your hand up" (General Practitioner 6) |

*(Continued)*

**Table 3.** (Continued)

| Sub-Theme | Experience of the CoP | Areas for Improvement | Participant Derived Solutions | Illustrative Quotes |
|---|---|---|---|---|
| Demand and Sustainability | High demand amongst community paediatricians | Sessions too frequent<br>After hours timing a barrier to attendance | Monthly rather than fortnightly<br>Flexible timing of sessions | *"I do hope that it can continue. . . having that connection with other people in this field is really good for me. . ." (Psychologist 12)*<br>*"A lot of the paediatricians. . . couldn't even get their names on the list. . . I know that amongst our practice of about 15 paediatricians, there was only myself and participant 10. . .and the interest in amongst our group was a lot of higher than that. . ." (Paediatrician 17)* |

There were differing views amongst clinicians about the delivery of the CoP via an online videoconferencing format. Many reported that virtual delivery enabled better attendance but had a negative impact on group dynamics. The after-hours timing (requested in the co-development sessions by GPs) was a significant barrier to attendance, with some citing competing commitments or fatigue at the end of the working day as key factors. Overall, clinicians requested greater flexibility regarding timing of sessions. Resources provided through the CoP, including session recordings, lecture slides, and case summaries, were universally well received. Clinicians reported that they supported self-directed learning and shared the resources with colleagues.

**Subtheme: Multidisciplinary model, group dynamics and facilitation.** Clinicians reported that the multi-disciplinary aspect of the CoP was felt to be extremely valuable, but also presented some challenges. Clinicians across all professional groups felt they gained important insights into the skill set, knowledge base, case load and difficulties faced by others working in the field of child mental health. Clinicians disclosed that the multi-disciplinary group also enabled them to expand their professional networks and obtain peer support. However, some clinicians thought the didactic teaching assumed gaps in knowledge, that were present for some but not for others. Possibly the multi-disciplinary nature of the model, while helpful in many respects, at times made it difficult to meet the specific learning needs of each clinician group.

Facilitation was described as an important mechanism to moderate group discussion and dynamics. Participants stated that, at times, inadequate management of the group dynamic led to unequal contributions from some individuals, or exclusion of some clinician groups. The group dynamics were also felt to be adversely impacted by technology and the absence of clear rules for online engagement. Most clinicians expressed frustration that others would contribute to discussions without introducing themselves or providing their professional background making it difficult to build group cohesion, develop connections and understand the context of comments made. Clinicians also reported that the videoconferencing format also made it difficult to perceive non-verbal communication cues and worked as a barrier to making spontaneous comments or asking questions. Some clinician suggestions included the use of smaller, online breakout groups to foster stronger group cohesion.

**Subtheme: Demand and sustainability.** Clinicians generally reported that their involvement in the CoP was worthwhile, and they looked forward to the sessions. Despite some challenges, many described the multi-disciplinary approach as timely, particularly for solo practitioners. A high demand for involvement was reported amongst paediatricians, who are currently the main prescribers of psychotropic medications for this patient population. Clinicians expressed a hope the CoP would continue, as it gave them a sense that teamwork was

**Table 4. Quotes representing theme 2: Change in clinical practice.**

| Sub-Theme | Examples | Illustrative Quote |
|---|---|---|
| Change in Knowledge Assessment and Management | Increased knowledge of pharmacological management of child mental health conditions<br>Assessment of medical instability in anorexia nervosa and criteria for admission<br>Risk assessment of self-harm/suicidality<br>Lowered threshold for trial of SSRI and titration to maximal dose | *"From talking to the other member of the Community of Practice, medication seems to be much more widely used. . . that I thought it was" (General Practitioner 19)*<br>*"We learned a little bit more about the assessment of the severity [of anorexia nervosa]. . . and when you need to sort of put them in hospital. . . blood pressure and heart rate. . . how much they have lost and how quickly. . . those specific assessment questions were quite helpful" (Counsellor 9)*<br>*"I take shortcuts in trying to come to a formulation, and I think what these Community of Practice sessions have done is made me re-evaluate. . . going back and being absolutely diligent. . . in a really thorough history" (Paediatrician 17)*<br>*"I think following the depression series I probably have been a bit more proactive in treating younger people for depression. . ." (General Practitioner 11)*<br>*". . . that access to Psychiatrist 1 and Psychiatrist 2 has just been immense because it's getting an answer. . . what am I going do for this patient this week to make a difference? Rather than, like I said, just stringing a patient on, um, and putting them on a long wait list of somebody else to, to sort it out. . . [it] gave me an instant solution, so that was good" (Paediatrician 15)* |
| Patient Advocacy and Communication | Promotes clinically relevant communication between clinicians<br>Improved communication with families | *"I feel like I've got a better understanding. . . if I get a referral. . . I really have a good understanding of their knowledge now. . . it actually has helped as far as you know reporting back. . . what needs to go in the reports" (Psychologist 16)*<br>*". . . to prepare [patients]. . . so that they can ask questions about the potential side effects and what the medication is for. . . I think lots of people feel disempowered around that. . ." (Psychologist 20)* |

possible in the future. Many felt that monthly sessions, rather than fortnightly would support more sustained engagement.

## Theme 2: Change in clinical practice

**Subtheme change in knowledge, assessment and management.** Table 4 summarises themes relevant to change in clinical practice. Clinicians described a broadening of clinical knowledge in several key areas of child and adolescent mental health, particularly the principles guiding pharmacological management. Discussions also challenged perceptions of prescribing psychotropic medications in younger patients. Participants provided positive feedback on the sessions which covered assessment and management of eating disorders and self-harm/suicidality. Secondary consultation was accessed outside of CoP sessions, by diverse clinician groups and was found to be associated with a significant, positive impact on patient management. Clinicians provided examples where access to psychiatry expertise expedited patient management, reduced referrals, provided additional management options, supported continuity of care and professional development.

## Subtheme: Patient advocacy and communication

Clinicians reported that the CoP promoted more clinically relevant communication between colleagues and that increased knowledge also translated to improvements in communication with families, patient advocacy, and care. This was often reported amongst psychologists, particularly with respect to medication management.

## Theme 3: Impact on clinician wellbeing

**Subtheme: Reduced feelings of isolation, stress and stigma.** Table 5 summarises themes relating to clinician wellbeing. Clinicians described how the multi-disciplinary model enabled them to form peer support networks, which lessened feelings of isolation and reduced the

**Table 5. Quotes representing theme 3: Impact on clinician wellbeing.**

| Sub-Theme | Examples | Illustrative Quote |
|---|---|---|
| Reduced Feelings of Isolation, Stress and Stigma | Formation of peer support networks Reduced stigma of struggling to manage challenging cases Secondary consultation alleviated stress | *"I think for a lot of us working in the community it can be, you know, depending on our roles, it can be a bit isolating at times, in terms of how much contact we have with other professionals. So that's probably the part that I liked the most about it. . ."* (Psychologist 8) *". . .. [it's] comforting to know that you're not the only one that struggles with really challenging clients"* (Counsellor 9) *". . . knowing that there was a consultant psychiatry service that we could ring for advice. . . if I get really stuck again, I have got a few other options to consider. . ."* (Paediatrician 5) *". . . it really wasn't clear to me. . . until I'd actually referred on. . . how much responsibility I was taking on. . . how much that was impacting me, even outside of work. . . just that mental space I didn't have. . . I was constantly worried and thinking about this person"* (Psychologist 8) |
| Increased Clinician Confidence | Increase in clinician confidence with improvement of knowledge and validation of clinical practice | *". . . since getting to know [Psychiatrist 1] better through, um, the Community of Practice, I've been much more open to the idea of managing my own patients"* (Paediatrician 17) *" I think it's probably reassured me that when my complexity radar goes off, that that's real and to trust it. Because everybody's radar would have gone off in a similar way. . ."* (Psychologist 18) |

stigma of struggling to manage challenging patient presentations. Medical practitioners working in a private practice setting expressed that this aspect of the CoP was highly valued. The CoP helped some clinicians reflect on the impact of challenging patients on their own mental health. In addition, the CoP provided solutions through increasing awareness of services available for complex case management. Participants reported that this promoted a sense of shared responsibility for patient care and helped to manage clinician stress and burnout.

**Subtheme: Increased clinician confidence.** Most clinicians, across all professional groups, reported an increase in confidence associated with validation of their current clinical practices. Clinician confidence also increased with acquisition of knowledge and access to support via secondary consultation. This was noted particularly amongst paediatricians.

**Secondary consultation service.** Over a two-week logbook maintained by one of the child psychiatrists, 22 clinicians (5 GPs and 17 paediatricians), accessed the consultation service. Consultations included phone (n = 9), email (n = 3) and in-clinic appointments (n = 10) requesting medication (n = 12) and diagnostic advice (n = 10). All consultations resulted in the patient being referred back to the clinicians therefore avoiding potential further referrals to the tertiary Child and Adolescent Mental Health Service (CAMHS).

## Discussion

We aimed to determine whether the COMPASS model was associated with increased community clinician self-reported confidence in: Managing common child and adolescent mental health disorders; navigating the mental health care system; diagnosing conditions; prescribing

psychotropic medications as well as the impact on clinician practice and wellbeing and outcomes of patients referred by COMPASS clinicians to the child psychiatrist. We found increased community clinician confidence in diagnosis and management of common paediatric MH conditions, how to navigate the mental health care system, and reduced need for ongoing tertiary mental health care. Overall, benefits were greatest for non-pharmacological management of self-harm and suicidal ideation in both child and adolescent mental health conditions. These findings were reflected in the qualitative analysis, with clinicians providing numerous examples where content from the CoP sessions was used to guide patient management and care. Furthermore, the qualitative analysis found that the COMPASS model improved communication with colleagues, patient advocacy, supported clinician wellbeing and reduced feelings of professional isolation and burnout. This is likely to improve clinical care for paediatric patients.

Whilst there were many positive outcomes of COMPASS, the multi-disciplinary aspect of this model came with challenges which need to be anticipated and planned for, if the model is to function well in future iterations. The importance of effective facilitation in managing group dynamics is key, and benefits from including facilitators with clinical experience and from a variety of professional backgrounds. Groups should also have an equal representation of participants from various clinical disciplines to ensure greater cohesion. A co-design process, along with frequent feedback and structured teaching cases may ensure a more inclusive curriculum, so that the learning needs are met for all professional groups. The aims of COMPASS, as a non-hierarchical, shared learning experience should be clearly articulated, so that participants set appropriate expectations around their involvement. Effective use of the video-conferencing platforms, such as breakout rooms, the chat function, and guidelines for online engagement, may encourage greater participation across clinician groups. Future COMPASS sessions may need to be offered both in person, and online to meet clinician preferences.

To our knowledge, this is the first such evaluation of an Australian model designed to upskill community clinicians. Whilst previous models have been implemented to upskill clinicians in paediatric mental health, few have been evaluated. The MCPAP allows enrolled primary care providers to get assistance for children in their care [28] including in-person psychiatric or clinical assessment, transitional therapy, and/or facilitated linkage to community resources. The program does not directly upskill community clinicians however, a retrospective analysis of survey data from primary care clinicians utilising MCPAP over 3.5 years, found a greater proportion of clinicians were able to meet the mental health needs of their patients (from 8% to 63%), suggesting some indirect upskilling of clinicians [22].

There is evidence to support the use of CoP models to upskill community clinicians in providing mental health care for children. Project TEACH is a pediatric specific, community-based intervention that shares key components of the COMPASS model: secondary consultation from tertiary specialists and structured education [29,30]. However, this program was targeted specifically to primary care practitioners, and was not truly multi-disciplinary in nature. Project TEACH was delivered to 139 primary care clinicians in New York, and included four 3-hour evening core educational sessions, access to secondary telephone consultation and referral support [29]. Results of a qualitative study (n = 30), found that clinicians reported greater confidence communicating with families, assessing severity of mental health presentations, prescribing medication, and developing management plans [29]. Participation in Project TEACH was also reported to strengthen relationships with MH specialists, reduce barriers to seeking advice and generally helped them work with the mental health system in a more integrated manner [26]. The multi-disciplinary nature of COMPASS meant that professional connections were strengthened across several clinician groups: primary care, paediatrics, psychology and psychiatry. As such the COMPASS model is unique, in being able to amplify

the benefits of inter-disciplinary learning and collaboration to support child and adolescent mental health care.

Another point of difference for our COMPASS model was that of co-design. One the major barriers to participation highlighted by participants involved in Project TEACH was the time and travel commitment involved, as sessions were longer and delivered in person. Co-design was a unique aspect of our COMPASS model, which overcame such barriers for busy clinicians who requested shorter, more frequent sessions which were scheduled outside business hours.

Like Project TEACH, BHIP comprises a structured education program and on demand psychiatry consultation for primary care clinicians [30,31]. However, in contrast to COMPASS, BHIP is a more intensive and costly program, embedded as a clinical service in the primary care practice, supported triaging of referrals and including clinicians who deliver brief psychotherapy. BHIP also provides support for practice transformation including operational sessions focusing on clinical and business workflow, crisis protocols, care pathways and linkage to speciality services. A survey of participants (n = 66, response rate 81.5%) found 95% acquired new knowledge about psychotropic medication, psychological therapies and felt they were able to provide better patient care [30]. The results of these studies align with the COMPASS evaluation.

## Strengths and limitations

This mixed methods study has several strengths. Appropriate quantitative and qualitative methodologies were chosen to meet the aims of the evaluation and 86% completed pre and post surveys. Clinicians were from a range of disciplines with varying clinical experience suggesting our findings could generalise to a range of healthcare providers. However, clinicians self-selected into COMPASS and are thus likely to represent a group motivated to improve pediatric mental health care. They self-reported on confidence which may lead to response bias however, we note that not all changes in confidence were significant. As it was a pre-post design, causality cannot be assumed. Potential confounders such as clinicians accessing other MH training to improve confidence or organising their own access to a child psychiatrist cannot be ruled out. Future research could address this by conducting an adequately randomized controlled trial of COMPASS, whereby clinicians are randomly assigned to participate in COMPASS or not. However, it may prove difficult to recruit clinicians to a control group.

A purposive sample of clinicians completed qualitative interviews, including an even distribution of medical and non-medical professionals. The interview transcripts were double coded and data saturation was reached after analysing approximately half of all transcripts. Researchers undertook fortnightly meetings to resolve coding discrepancies and track coding decisions. They also completed journaling activities during the data collection and analysis phase to reflect on personal biases and preconceptions, as well as their relationship to participants, and how this may impact the research outcomes. Although we reached data saturation with our qualitative interviews, data saturation might not capture the full range of perspectives and experiences in the broader clinical community. Our sample was mostly females, who were relatively experienced clinicians. It may have been useful to hear from less experienced clinicians to better understand whether COMPASS met their needs. One child psychiatrist completed a brief snapshot of consultations and future research should aim to capture these outcomes over a longer period.

We have evaluated COMPASS in its first year of operation and from clinicians' perspectives only. Future research could evaluate the longer-term impacts on clinicians' confidence and wellbeing, impacts on patient outcomes, and barriers and enablers to scaling the model.

## Conclusion

The COMPASS model meets several strategic policy priorities outlined in Royal Commission into Victoria's Mental Health System Final Report and the National Children's Mental Health and Wellbeing Strategy [8,32]. Recommendations 58 and 64 from the Royal Commission seek to "advance workforce capabilities, professional development and drive innovation in mental health-care treatment and support"[8]. COMPASS provides a novel approach to meeting these objectives and could be expanded to areas of even greater need such as rural communities [10]. Furthermore, the multi-disciplinary nature of this model, along with access to child psychiatry consultation, meets Objective 2.2 in the National Children's Mental Health and Wellbeing Strategy which calls for better integration and coordination of services, as well as collaborative care [32]. Our evaluation of COMPASS shows this feasible and desirable. It is now in its third year of operation with the NWMPHN and is ripe for expansion nationally. It could be further scaled to other healthcare systems where there is a lack of access to child psychiatry expertise and a need to upskill community clinicians. If Australia is to meet the growing demand for paediatric mental health care, we need to upskill existing, community-based providers in a manner that is feasible and sustainable.

## Supporting information

**S1 Appendix. Participant flowchart.**
(PDF)

**S2 Appendix. Clinician pre survey.**
(PDF)

**S3 Appendix. Clinician post survey.**
(PDF)

**S4 Appendix. Participant information form.**
(PDF)

**S5 Appendix. Qualitative interview guide.**
(PDF)

**S6 Appendix. Secondary consultation record log.**
(PDF)

## Acknowledgments

The authors would like to acknowledge the North Western Melbourne Primary Health Network and The Royal Children's Hospital Mental Health for their invaluable partnership. The authors would like to especially thank the child psychiatrists who led the Community of Practice sessions and the clinicians who dedicated their time to complete surveys and provided valuable feedback.

## Author Contributions

**Conceptualization:** Sonia Khano.

**Writing – original draft:** Elise D'Abaco, Sonia Khano, Al Giles-Kaye, Jag Dhaliwal, Ric Haslam, Chidambaram Prakash, Harriet Hiscock.

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
