## [Decision Letter · Decision Letter 0]

7 Sep 2023

PONE-D-23-14369Impact of a collaborative model on community clinician confidence and competence in child and adolescent mental health: mixed method analysis.PLOS ONE

Dear Dr. Hiscock,

Thank you for submitting your manuscript to PLOS ONE. After careful consideration, we feel that it has merit but does not fully meet PLOS ONE’s publication criteria as it currently stands. Therefore, we invite you to submit a revised version of the manuscript that addresses the points raised during the review process.

Please follow the reviewers' comments to make the necessary adjustments for your manuscript to be considered for publication in our journal.

We look forward to receiving your revised manuscript.

Kind regards,

Lea Sacca

Academic Editor

PLOS ONE

Journal Requirements:

2. In the ethics statement in the Methods, you have specified that verbal consent was obtained. Please provide additional details regarding how this consent was documented and witnessed, and state whether this was approved by the IRB

5.We note you have included a table to which you do not refer in the text of your manuscript. Please ensure that you refer to Tables 4, 5 and 6 in your text; if accepted, production will need this reference to link the reader to the Table.

Reviewers' comments:

Reviewer's Responses to Questions

**Comments to the Author**

1. Is the manuscript technically sound, and do the data support the conclusions?

Reviewer #1: Yes

Reviewer #2: Partly

2. Has the statistical analysis been performed appropriately and rigorously? 

Reviewer #1: Yes

Reviewer #2: No

3. Have the authors made all data underlying the findings in their manuscript fully available?

Reviewer #1: Yes

Reviewer #2: Yes

4. Is the manuscript presented in an intelligible fashion and written in standard English?

Reviewer #1: Yes

Reviewer #2: Yes

5. Review Comments to the Author

Reviewer #1: Abstract

The abstract provides a concise overview of the study's objectives, methods, and anticipated results. However, there are several areas where it could be improved to enhance clarity and transparency.

1. Specificity of objectives: The objectives mentioned are quite general ("confidence, competence, and well-being of community clinicians"), and it might be beneficial to provide more specific details about what aspects of confidence, competence, and well-being are being measured. This would give reviewers a clearer understanding of the study's focus.

2. Lack of background: The abstract lacks background information about the current state of child and adolescent mental health care, the challenges faced by community clinicians, and the need for collaborative models like COMPASS. Providing a brief context would help reviewers understand the significance of the study.

3. Methods and measures: The abstract briefly mentions pre-post surveys and semi-structured interviews as methods, but it doesn't elaborate on the specific scales or tools used for assessing confidence, competence, and well-being. Including details about the measurement tools and their validity/reliability would enhance the methodological transparency.

4. Qualitative analysis approach: The description of the qualitative analysis is quite brief ("inductive approach" and "content analysis"). Elaborating on the process of qualitative data analysis, including coding strategies, inter-coder reliability, and how themes are being generated, would provide a clearer picture of the rigor in the qualitative component.

5. Outcomes of consultations: The abstract mentions a child psychiatrist's snapshot of consultations without explaining the purpose of this snapshot or how it contributes to the study's objectives. Clarifying how this data will be used and what insights it aims to provide would be beneficial.

Introduction

The introduction provided seems to address the research topic of child and adolescent mental health and the need for improved support for community-based clinicians. However, from a peer-review perspective, there are several limitations and areas for improvement that should be considered before publication:

1. Clarity and focus: While the introduction covers a wide range of relevant information, it could benefit from greater focus. The introduction should clearly state the research problem, objectives, and research questions. The reader should be able to understand the main purpose of the study from the introduction alone.

2. Citation usage: The introduction relies heavily on citations, which is good for providing evidence but can become overwhelming. It's important to strike a balance between providing evidence and maintaining a coherent narrative. Some information might be better summarized instead of directly quoting from sources.

3. Transition and flow: The introduction lacks smooth transitions between the different sections. Each paragraph should naturally lead to the next, guiding the reader through the argument. Consider using transitional phrases to connect ideas and improve the overall flow.

4. Research gap: While the introduction mentions the lack of evaluated models in Australia, it could further emphasize the research gap in the existing literature. What specific aspects of the current knowledge are insufficient? How will the proposed study address these gaps?

5. Theoretical framework: It would be beneficial to include a brief overview of the theoretical framework that underpins the study. What theoretical perspectives are guiding the research? This will help readers understand the context within which the study is situated.

6. Hypotheses/research questions: The introduction should clearly present the hypotheses or research questions that the study aims to address. This provides readers with a clear roadmap for what the study intends to investigate.

7. Significance and implications: Highlight the significance of the study in addressing the identified gaps. What are the potential implications of the study's findings for the field of child and adolescent mental health, clinical practice, and policy development?

8. Methodology preview: While the introduction does briefly mention the study's implementation and evaluation of the COMPASS model, a clearer preview of the research methodology would be valuable. This could include a brief overview of the study design, data collection methods, and analytical approaches.

9. Engagement with existing literature: While the introduction references a few statistics and the lack of evaluated models, it could engage more deeply with the existing literature on community-based mental health care and the challenges faced by clinicians. This will help position the study within the broader academic discourse.

10. Conciseness: Some sections of the introduction could be condensed without losing the essential information. This will help maintain reader engagement and prevent information overload.

Methods/materials

The provided Methods section outlines the implementation of the COMPASS model and the research methods used to assess its impact. However, there are areas within this section that could be improved from a peer-review perspective:

1. Clarity of procedures: While the description of the COMPASS model and its implementation is quite detailed, there could be clearer organization and separation of the different stages or components. The description could be broken down into subsections, such as "Recruitment of Participants," "Co-Development of CoP," "Delivery of CoP Sessions," "Qualitative Interviews," etc., to enhance readability and structure.

2. Rationale for method selection: Provide more justification for the chosen methods. Why were CoP sessions and qualitative interviews selected as the main approaches to evaluate the COMPASS model's impact? How do these methods align with the study's objectives and research questions?

3. Sampling and recruitment details: Provide more information about the size and characteristics of the participant pool. How many clinicians were initially approached, and how many participated in the CoP sessions and surveys? How were participants selected for interviews? Including this information provides context for the representativeness of the sample.

4. CoP content and structure: While the content of the CoP sessions is mentioned, there could be more detail about the development process, how the topics were selected, and the rationale behind choosing those specific areas. This could help readers understand the design and intent of the CoP sessions better.

5. Ethical considerations and informed consent: While the ethics approval is mentioned, it would be beneficial to elaborate on the ethical considerations taken into account during the study. Describe the steps taken to ensure participant confidentiality and how informed consent was obtained, particularly for both surveys and interviews.

6. Data collection and analysis: Elaborate on the process of data collection, management, and analysis. How were surveys administered? How were survey responses coded and analyzed? Similarly, how were qualitative interviews conducted, transcribed, and analyzed?

7. Validity and reliability: Discuss steps taken to ensure the validity and reliability of both the quantitative and qualitative data. This could include measures to ensure consistent administration of surveys, inter-rater reliability for coding qualitative data, and methods to enhance data triangulation.

8. Limitations and potential biases: Address potential limitations and biases in the study design and methods. For instance, what are the limitations of relying on self-reported measures of confidence? How might the use of a single child psychiatrist influence the consultation service's outcomes? These can also be included in the Discussion.

9. Alignment with research objectives: Continuously relate the methods back to the research objectives and questions. Ensure that each method's purpose is clear and directly connected to the study's overall aims.

10. Appendices: References to appendices are made without providing the content of the appendices. While they should be provided separately, consider adding a brief description of what each appendix contains to guide readers.

Results

The Results section of the study appears to provide a comprehensive overview of the findings and outcomes of the COMPASS Online CoP intervention. However, there are a few areas where improvements could be made:

1. Clarity and structure: While the section is well-organized and divided into themes and subthemes, some of the subthemes could benefit from clearer headings or labels. This would help readers quickly understand the main points of each subtheme and how they relate to the overall theme.

2. Data representation: The use of tables and figures to present quantitative data is helpful, but ensure that the data are clearly labeled and explained in the text. For instance, in Table 1, it would be helpful to include a brief description of each characteristic being presented, especially for readers who may not be familiar with the field.

3. Quotation usage: The use of quotations from participants to support themes is valuable, but consider providing more context for each quote. Briefly explain who the participant is (e.g., GP, psychologist) and their role, which would add credibility to the qualitative findings.

4. Statistical significance and effect size: When presenting quantitative data, it's important to indicate whether any observed changes are statistically significant and, when possible, provide effect size information. This helps readers understand the magnitude of the changes and their significance.

Discussion and conclusion

While the content of the discussion covers relevant aspects and findings of the study, there are some limitations and areas that could be improved before publication. Here are some suggestions and potential limitations to consider:

1. Clear linkage to research objectives and findings: The discussion should start by explicitly linking back to the research objectives or hypotheses stated in the introduction. This helps readers understand how the study addressed the research questions and what the key findings were.

2. Causality and study design: It's important to emphasize that the study used a pre-post design, which allows for examining changes within the same group over time but doesn't establish causality. This limitation should be acknowledged, and potential confounding factors discussed. Additionally, the discussion could highlight how future research could incorporate control groups or more rigorous study designs to strengthen causal inferences.

3. Comparative analysis with other models: While the discussion mentions other models like Project TEACH and BHIP, it could provide a more detailed comparative analysis of COMPASS with these models in terms of outcomes, implementation, and potential advantages and disadvantages. This would help readers understand how COMPASS contributes uniquely to the field.

4. Challenges and lessons learned: Beyond facilitation and group dynamics, other challenges that emerged during the implementation of the COMPASS model should be discussed. This could include issues related to participant engagement, technology, or any unforeseen barriers that affected the model's effectiveness.

5. Generalizability and external validity: Discuss the generalizability of the findings. How representative is the sample of the broader clinician population? Are there potential biases in the sample that could affect the applicability of the findings to other settings or regions?

6. Implications for policy and practice: In addition to addressing the strategic policy priorities mentioned, discuss how the findings of this study could inform decision-making at both policy and practice levels. How could the COMPASS model be adapted or replicated in other regions or healthcare systems?

7. Limitations of qualitative analysis: The qualitative analysis reached data saturation after analyzing about half of the transcripts. While this is common in qualitative research, acknowledge that data saturation might not capture the full range of perspectives and experiences. Discuss potential areas that might have been missed due to this limitation.

8. Future Directions: Provide concrete suggestions for future research directions. This could include exploring the long-term sustainability of the COMPASS model, evaluating its impact on patient outcomes, or investigating the barriers to wider implementation.

9. Language and style: Ensure that the language in the discussion is precise and concise. Avoid overly technical jargon, and explain complex concepts or findings in a clear and accessible manner.

Reviewer #2: General Comments:

1. The manuscript presents both quantitative and qualitative evaluation of the COMPASS model (mixed method analysis), on a highly relevant topic.

2. The title suggests the idea of different methods of analysis, yet the statistical analysis is not clear and lacks innovation

3. The main objective of the study should be clearly defined. It was confusing to understand precisely what the study aimed to evaluate.

4. The description of the study's methodology and the statistical analysis should be improved.

5. The methodology and results sections need restructuring.

6. It's not clear whether the COMPASS model is an approach created by the study or an existing program.

Title: Impact of a collaborative model on community clinician confidence and competence in child and adolescent mental health: mixed method analysis

- Why the expression "confidence and competence"? Maybe describe it better in the methodology, or adjust it, as it becomes confusing when reading the title.

- I'm not sure if "mixed method analysis" is the best to use, because in terms of statistical methods, the study is limited. It's a study where I work with qualitative and quantitative data

Abstract:

- Objectives are extensive. Determine a primary objective.

- Results: Present the outcome of the association mentioned in the conclusion.

Introduction:

- It's extensive and still doesn't clarify well how the Community of Practice functions. Is it a government program, an initiative by someone, funded by someone? Does it still exist, or was it an emergency response during the pandemic? Is COMPASS an example of a Community of Practice? It all appears somewhat confusing.

- The study's objective is mixed with the COMPASS objective. Adjust it.

Methods:

- Who facilitated the sessions, a trained researcher, a clinical member?

- What were the development sessions like?

- Please include the ethical approval at the end of the methodology.

- Regarding ethical considerations, shouldn't a consent form have been signed? Additionally, for data collected from children and adolescents, shouldn't their guardians have been informed and given consent to participate? If it's secondary data, wouldn't there need to be a specific protocol for secondary data?

- Are the data from children and adolescents sourced from a database, medical record collection, or patients treated by the recruited clinicians?

- Reorganize the methodology (Design and Sample, Data Collection, Ethical Aspects...)

- Page 10, line 156: "Clinician pre and post surveys included demographic items (pre-survey only)." What data?

- Was a sample calculation performed? Is the sample size sufficient to make any claims?

- The methodology needs to be rewritten and reorganized. It requires information regarding the study design, the data used, and how they were analyzed (including specific statistical tests). I understand that COMPASS is an approach created by the study, so the objective and details of COMPASS should be included in the methodology.

-I suggest using a flowchart to describe participant recruitment and attrition.

Results:

- Page 12, lines 204-211: This belongs in the methodology.

- Tables: Describe all acronyms and symbols used in the captions, even if they are obvious (such as n).

- Use lowercase n for sample size. This applies to all figures and tables.

- Table 3: Describe statistical significance (*) in the caption.

- In the main text, focus more on describing what was statistically significant or not, rather than presenting only descriptive data.

- Figures 1 and 2: Remove the underlining from the figure titles.

- Pages 19-20, lines 269-281: This is methodology.

- Are the 27 eligible clinicians from the quantitative data among the 59 participants from the quantitative part of the CoP session data, or do you have two different samples?

Discussion:

- Page 415: "As per our aims, the COMPASS model was associated with increased community clinician knowledge of services, confidence in diagnosis and management of common paediatric MH conditions, and reduced referrals to CAMHS." Where is this data in the results?

- Describe the study's limitations more comprehensively.

Conclusion:

- Much of the conclusion comprises final considerations that should be in the discussion. The conclusion, in general, should be concise and address the objective and title.

6. PLOS authors have the option to publish the peer review history of their article (what does this mean?). If published, this will include your full peer review and any attached files.

Reviewer #1: **Yes: **Amanuel Abajobir

Reviewer #2: No

---

## [Author Response · Author response to Decision Letter 0]

24 Oct 2023

10th October 2023

Lea Sacca

Academic editor

PLOS ONE

Dear Lea

Re: PONE-D-23-1439

Many thanks for the opportunity to revise our paper which we believe is considerably improved thanks to reviewer comments.

We have now clearly stated our key aims (6 aims in all) for this paper and ensured that our aims align with the methodology and results. In doing this, we have deleted a small section of the original paper that focused on clinician self-reported factors that may affect their decision to refer a patient (summarized in the original table 2). We felt that these 4 factors were not particularly informative and lead to unnecessary lengthening of the paper, but we would be happy to re-instate them if you wish.

Kind regards,

Professor Harriet Hiscock MB BS, FRACP, MD, FAAHMS, GAICD

On behalf of the co-authors

Please see "PONE-D-23-1439- Response to Reviewers.docx" for full response to reviewers comments.

---

## [Decision Letter · Decision Letter 1]

16 Aug 2024

PONE-D-23-14369R1Impact of a collaborative model on community clinician confidence in child and adolescent mental health care, wellbeing, and access to child psychiatry expertise.PLOS ONE

Dear Dr. Hiscock,

 This paper was allocated to me as Academic Editor on 24 July 2024. My sincere apologies for the lengthy delay prior to this and I am not clear on the reasons for this. I am satisfied that your revisions have addressed the concerns of the previous reviewers. In addition, a third reviewer has also indicated that the revised paper can be accepted. I would, however, like a couple of minor revisions (see below) before I can recommend acceptance.  We invite you to submit a revised version of the manuscript that addresses the points below as soon as possible. 

ACADEMIC EDITOR:1. In your abstract, please replace your first heading "Objectives" with "Background".

2. The overall aim of your paper is to 'determine the impact of a collaborative model on community clinician confidence in child and adolescent mental health care' but what you are describing as Aims 1-6 are actually objectives within this broad aim. Please replace all the references to Aims 1-6 throughout the paper and replace these with Objectives 1-6.

We look forward to receiving your revised manuscript.

Kind regards,

Saiendhra Vasudevan Moodley, PhD

Academic Editor

PLOS ONE

Journal Requirements:

Reviewers' comments:

Reviewer's Responses to Questions

**Comments to the Author**

1. If the authors have adequately addressed your comments raised in a previous round of review and you feel that this manuscript is now acceptable for publication, you may indicate that here to bypass the “Comments to the Author” section, enter your conflict of interest statement in the “Confidential to Editor” section, and submit your "Accept" recommendation.

Reviewer #3: All comments have been addressed

2. Is the manuscript technically sound, and do the data support the conclusions?

Reviewer #3: Yes

3. Has the statistical analysis been performed appropriately and rigorously? 

Reviewer #3: Yes

4. Have the authors made all data underlying the findings in their manuscript fully available?

Reviewer #3: Yes

5. Is the manuscript presented in an intelligible fashion and written in standard English?

Reviewer #3: Yes

6. Review Comments to the Author

Reviewer #3: I think at present the paper is good to go. However there are some typos of capitatization etc which need to be corrected

7. PLOS authors have the option to publish the peer review history of their article (what does this mean?). If published, this will include your full peer review and any attached files.

Reviewer #3: No

---

## [Author Response · Author response to Decision Letter 1]

27 Aug 2024

Dear Dr Moodley,

Re: PONE-D-23-14369R1

Impact of a collaborative model on community clinician confidence in child and adolescent mental health care, wellbeing, and access to child psychiatry expertise.

I have made your requested changes to our revised manuscript i.e. to: 

1. In abstract, please replace your first heading "Objectives" with "Background".

2. Replace all the references to Aims 1-6 throughout the paper and replace these with Objectives 1-6.

Thank you for finally moving this paper on!

Kind regards,

Professor Harriet Hiscock

---

## [Editor Report · Decision Letter 2]

30 Aug 2024

Impact of a collaborative model on community clinician confidence in child and adolescent mental health care, wellbeing, and access to child psychiatry expertise.

PONE-D-23-14369R2

Dear Dr. Hiscock

We’re pleased to inform you that your manuscript has been judged scientifically suitable for publication and will be formally accepted for publication once it meets all outstanding technical requirements.

Kind regards,

Saiendhra Vasudevan Moodley, PhD

Academic Editor

PLOS ONE

---

## [Editor Report · Acceptance letter]

13 Sep 2024

PONE-D-23-14369R2 

PLOS ONE

Dear Dr. Hiscock, 

I'm pleased to inform you that your manuscript has been deemed suitable for publication in PLOS ONE. Congratulations! Your manuscript is now being handed over to our production team.

Kind regards, 

on behalf of

Dr. Saiendhra Vasudevan Moodley 

Academic Editor

PLOS ONE